# Generalist Predators Shape Biotic Resistance along a Tropical Island Chain

**DOI:** 10.3390/plants12183304

**Published:** 2023-09-18

**Authors:** Kris A. G. Wyckhuys, Johanna Audrey Leatemia, Muhammad Zainal Fanani, Michael J. Furlong, Baogen Gu, Buyung Asmara Ratna Hadi, Jeffij Virgowat Hasinu, Maria C. Melo, Saartje Helena Noya, Aunu Rauf, Johanna Taribuka, Yubak Dhoj Gc

**Affiliations:** 1Chrysalis Consulting, Danang 50000, Vietnam; 2Institute for Plant Protection, China Academy of Agricultural Sciences (CAAS), Beijing 100193, China; 3School of the Environment, University of Queensland, Saint Lucia, QLD 4067, Australia; m.furlong@uq.edu.au (M.J.F.); m.melo@uq.edu.au (M.C.M.); 4Department of Agrotechnology, Faculty of Agriculture, Universitas Pattimura, Ambon 97233, Indonesia; jaleatemia@hotmail.com (J.A.L.); jeffijhasinu@gmail.com (J.V.H.); saartjenoya314719@gmail.com (S.H.N.); anontari@yahoo.com (J.T.); 5Department of Agrotechnology, Faculty of Agriculture, Universitas Djuanda, Jl. Tol Jagorawi No 1, Ciawi, Bogor 16720, West Java, Indonesia; muhammaduzainale@gmail.com; 6Department of Plant Protection, Faculty of Agriculture, Institut Pertanian Bogor, Jl. Kamper Kampus IPB Dramaga, Bogor 16680, West Java, Indonesia; aunu@indo.net.id; 7Food and Agriculture Organization (FAO), 00153 Rome, Italy; baogen.gu@fao.org (B.G.); buyung.hadi@fao.org (B.A.R.H.); 8Food and Agriculture Organization (FAO), Bangkok 10200, Thailand; yubak.gc@fao.org

**Keywords:** island biogeography, global change, biodiversity loss, biological control, ecological intensification, invasion biology

## Abstract

Islands offer exclusive prisms for an experimental investigation of biodiversity x ecosystem function interplay. Given that species in upper trophic layers, e.g., arthropod predators, experience a comparative disadvantage on small, isolated islands, such settings can help to clarify how predation features within biotic resistance equations. Here, we use observational and manipulative studies on a chain of nine Indonesian islands to quantify predator-mediated biotic resistance against the cassava mealybug *Phenacoccus manihoti* (Homoptera: Pseudococcidae) and the fall armyworm *Spodoptera frugiperda* (Lepidoptera: Noctuidae). Across island settings, a diverse set of generalist lacewing, spider and ladybeetle predators aggregates on *P. manihoti* infested plants, attaining max. (field-level) abundance levels of 1.0, 8.0 and 3.2 individuals per plant, respectively. Though biotic resistance—as imperfectly defined by a predator/prey ratio index—exhibits no inter-island differences, *P. manihoti* population regulation is primarily provided through an introduced monophagous parasitoid. Meanwhile, resident predators, such as soil-dwelling ants, inflict apparent mortality rates up to 100% for various *S. frugiperda* life stages, which translates into a 13- to 800-fold lower *S. frugiperda* survivorship on small versus large islands. While biotic resistance against *S. frugiperda* is ubiquitous along the island chain, its magnitude differs between island contexts, seasons and ecological realms, i.e., plant canopy vs. soil surface. Hence, under our experimental context, generalist predators determine biotic resistance and exert important levels of mortality even in biodiversity-poor settings. Given the rapid pace of biodiversity loss and alien species accumulation globally, their active conservation in farmland settings (e.g., through pesticide phasedown) is pivotal to ensuring the overall resilience of production ecosystems.

## 1. Introduction

Islands are Nature’s test tubes, offering comparatively simple systems for experimentation with often numerous replicates [1]. As early as the Victorian era, their unique fauna and flora inspired naturalists and biographers, and gave rise to groundbreaking theories on species evolution or ecology [2]. Given that each island offers a unique complement of plant and animal species, as defined by its size and geographical isolation [3], they can help to assess the extent to which biodiversity and biostructure (i.e., species interactions) modulate ecosystem functioning. Decades of empirical study have shown that island area is a reliable proxy of species richness and various ecosystem-level properties, and that islands constitute ideal loci for experimental work. Indeed, island-area effects do not exclusively relate to insular communities, but to the functioning of ecosystems worldwide [4]. 

Global change is typified by marked species gains (i.e., invasive biota) and losses [5,6], and island ecosystems can help to anticipate their ecological consequences. More so, as global change dynamics are exacerbated in island settings, they serve as microcosms for the emerging landscapes of the Anthropocene [7]. For instance, island ecosystems are disproportionately affected by invasive species as they harbor comparatively more alien biota than mainland settings—a condition that may be foretelling of future ecosystems [8]. Equally, in small islands, the shorter food chains with few species and uneven presence of generalists offer impressions of a future under continuous defaunation or trophic downgrading, i.e., the disappearance of apex predators [9,10]. As such, island habitats pick up the ecological surprises that emanate from species invasion, predator loss or (pesticide-induced) disturbance before these become apparent in ‘mainland’ settings. Further, each island offers a ‘time capsule’ for a less biodiverse planet. As per the equilibrium theory of island biogeography (ETIB) [3], a given island marks a point along a biodiversity continuum to which (invasive) biota are periodically added or subtracted. To capture the ensuing biodiversity x ecosystem function interplay, islands offer myriad advantages over conventional (field, laboratory) experimental set-ups, as they allow for a full accounting of species interactions. Even though ‘static’ experiments have shown how high-diversity settings benefit from more stable ecosystem functions and services [11], particularly under stressful conditions [12], they often treat species as stand-alone entities instead of members of dynamic, interconnected communities. By doing so, those experiments discount the rewiring of interaction networks that follow a sudden apparition of new biota [13,14]. Thus, island settings can advance our mechanistic understanding of how biodiversity losses or gains shape ecosystem services [15], and offer insights that are particularly valuable under fast-progressing global change. 

Predators exert major impacts on ecosystems—often disproportionate to their size and/or abundance [16,17]—though are under strong anthropogenic pressure [9]. These patterns are pronounced in farmland ecosystems, where the richness of predators and other so-called natural enemies, in addition to their abundance, supports ecosystem services such as biological pest control [18]. Their long-term impact even surpasses that of synthetic insecticides [19]. Even though biological control is valued at US $100 billion/year in the global agroecosystem [20], natural enemies are negatively affected by pesticide applications, landscape simplification and habitat loss. As a result, insect predator numbers are declining more rapidly than many other trophic guilds [21], which has repercussions for biological control and ecosystem resistance against invasive species within and beyond farm borders. Indeed, (generalist, opportunistic) predators provide so-called ‘biotic resistance’ against invasive biota, as demonstrated for insectivorous birds [22], granivorous crickets [23] or ants [24,25]. Building upon Ohm’s law, Chapman [26] first defined ecosystem resistance as the capacity of the environment to prevent a given species from attaining its full potential. This concept was further refined by Elton [27] and ultimately gave rise to the biotic resistance hypothesis, which posits that less biodiverse ecosystems are more vulnerable to invasion than more biodiverse ones. Though this is supported by theoretical studies, results of empirical work are decidedly mixed and often muddled by co-varying environmental factors [28]. Irrespective of the above, low-diversity settings with species-poor invertebrate predator communities are often more susceptible to pest invasion [29]. As a consequence, it has been theorized that exotic species are more likely to establish and exert damaging impacts on islands as compared to continents. However, as supporting evidence for this hypothesis is weak [30], one cannot directly assume that island communities are intrinsically less ‘resistant’ to invasion by virtue of their reduced richness of, e.g., invertebrate predator species [31]. 

Over the past decade, Indonesia’s island settings have been invaded by two notorious agricultural pests. First, the Neotropical cassava mealybug *Phenacoccus manihoti* Matile-Ferrero (CAM; Hemiptera: Pseudococcidae) made its arrival in western Java in 2010 and subsequently spread across Indonesia’s island chain through natural and human-assisted means [32]. During the initial stages of its invasion, *P. manihoti* attained infestation levels of several hundreds of mealybug individuals per cassava tip and lowered root yields by up to 75% [33,34]. Second, the fall armyworm *Spodoptera frugiperda* Smith (FAW; Lepidoptera: Noctuidae) invaded Indonesia in early 2019 after a devastating passage through Africa, the Middle East and South Asia [35]. FAW primarily affects maize where it reduces yields by an estimated 17%—as experimentally derived—across its native and invasive range [36], and now annually inflicts US $9.4 billion losses in Africa alone [37]. A diverse suite of arthropod predators is associated with *P. manihoti* and *S. frugiperda* in their invaded range [38,39]; resident natural enemies that attack phylogenetically related native biota likely exert the strongest impacts during the initial invasion stages [40]. Yet, at those times when invasive biota attain high-infestation levels, generalist predators do not necessarily exert sufficiently strong top-down control to provide effective population regulation, as demonstrated for the gypsy moth and winter moth in the USA [41,42]. Predator responses are also strongly context-specific: in certain (more or less biodiverse, disturbed or pristine) settings, resident predator communities can be rapidly saturated by food availability [42]. When gauging the outcomes in terms of invasion success, both predator occurrence and local (agro-)ecological context, e.g., island size, are thus important [43]. So far, it is unknown to what extent resident predator communities provide biotic resistance against either CAM or FAW, and how such is modulated by the community composition, biodiversity or broader ecosystem health of a given locale. Hence, irrespective of the contribution of biotic resistance to invasive species management, biological control science does not necessarily feed into invasion ecology research, even for some of the world’s top invasive crop pests [44]. 

In this study, we build upon earlier work to illuminate islands’ potential to inform biological control practice [45]. Drawing upon a set of observational and manipulative assays conducted across nine Indonesian islands, we investigate whether species richness (as tentatively defined by island size) affects predator recruitment in pest-invaded crops (for CAM, over 2017–2018) and mortality of the various developmental stages of an invasive herbivore (for FAW, during 2022). For CAM specifically, island sites are also differentially affected by presence of the exotic *Anagyrus lopezi* De Santis (Hymenoptera: Encyrtidae), a specialist, i.e., monophagous parasitoid of CAM that was introduced into Indonesia in late 2014, providing effective biological control [33]. Our study provides an unique assessment of the extent to which island size affects predator-mediated biotic resistance against two invasive crop pests. Though exploratory in nature, our study informs biological control science and practice [46], but equally extends its reach into the invasion ecology domain and lends support to non-chemical means of invasive species mitigation. 

## 2. Materials and Methods

### 2.1. CAM Natural Enemy Surveys

Over the course of 2017–2018, insect surveys were carried out in 77 CAM-invaded cassava fields across five large Indonesian islands (Appendix A). From large to small in size, surveyed islands comprised Sumatra (*n* = 11 fields), Java (*n* = 36), Timor (*n* = 10), Flores (*n* = 11) and Lombok (*n* = 9). Fields were managed by individual farmers using locally prevailing practices; as cassava is a low-value, long-season crop that is primarily cultivated by resource-poor farmers, (especially) in Flores, Timor and Lombok, where pesticide use was rare. Survey protocols were adopted from Wyckhuys et al. [34]. In brief, we selected older fields (i.e., 8–10 months of age) in the main cassava-growing districts of each island with individual fields spaced at min. 1 km distances. Fields were 1.52 ± 0.17 ha in size (mean ± SE) and planted with one out of five locally popular varieties. Field visits were planned during the local dry season; most fields were visited during October–November 2017, while an additional set of fields in Lampung (Sumatra) and eastern Java were also sampled during September–October 2018. In each field, five linear transects were randomly chosen and ten plants were sampled along each transect. As such, for a total of 50 cassava plants per field, we visually recorded the number of *P. manihoti* individuals, the prevailing associated predators, i.e., lacewings (Neuroptera: Chrysopidae), ladybeetles (Coleoptera: Coccinellidae) and spiders (Araneae). In addition, considering how mealybug-tending ants interfere with CAM predation [47], we recorded the number of individual ants per cassava plant. In-field identification of *P. manihoti* was based on morphological characteristics such as coloration, size and absence of abdominal waxy filaments. Following transect walks, we computed the average abundance of *P. manihoti* and its associated predators or tending ants per plant. Field locations were recorded using a handheld GPS unit (Garmin Ltd., Olathe, KS, USA), while voucher specimens of *P. manihoti* and *A. lopezi* were deposited at Bogor Agricultural University (Bogor, Indonesia). A predator/prey ratio was further computed by dividing the field-level abundance of all predators by that of *P. manihoti*, i.e., specifically geared to this single invasive pest. Predator/prey ratio is a sound proxy of biological control activity or predation pressure [48], and might thus be indicative of biotic resistance. 

Given that *A. lopezi* had been successfully established in Sumatra, Java and Flores by late 2017, we equally ascertained its establishment and parasitism rates in each of the surveyed fields. In brief, 20 mealybug-infested cassava tips were collected from each cassava field and transferred to a laboratory to assess parasitoid emergence and identity [34]. Upon arrival in the laboratory, each tip was carefully examined, the total number of *P. manihoti* was counted, and all (insect) predators were removed. Cassava tips were then placed singly into transparent polyvinyl chloride (PVC) containers, which were closed with fine cotton fabric mesh. Over the span of three weeks, PVC containers were inspected daily for emerged parasitoids, and *A. lopezi* parasitism levels were recorded. Parasitism levels by locally occurring parasitoids were recorded, even though those were not identified to species levels. 

In our analyses, we considered both field-level average abundance data and plant-level data, i.e., those recorded at each of the 50 individual cassava plants surveyed in a given sampling bout. Prior to analysis, all data were checked for normality and homoscedasticity. Normally distributed data were subject to parametric tests while those that did not meet normality assumptions were analyzed using a non-parametric Kruskal–Wallis test. When analyzing output of multiple pair-wise Kruskal–Wallis tests, significance values were adjusted using the Bonferroni correction. As the introduction of the *A. lopezi* parasitoid impacted mealybug abundance, *A. lopezi*-colonized fields were analyzed separately from fields in which this parasitoid was absent. For either set of fields, we carried out a regression analysis to relate the density of *P. manihoti* to that of the entire predator community and the respective predator/prey ratio. Further, we used one-way analysis of variance (ANOVA) to compare (field-level) predator or mealybug abundance between islands of varying size. Lastly, a stepwise multiple regression analysis was performed to assess the extent to which predator abundance or predator/prey ratio are determined by island, CAM and non-CAM mealybug abundance or *A. lopezi* parasitism level. All statistical analyses were conducted using SPSS.

### 2.2. FAW Predation Trials 

A second set of experiments was carried out over the course of 2022 in Indonesia’s Maluku province, i.e., an area that was recently invaded by FAW. Experimental work was conducted in 4 different islands of varying size and inter-island distances (Appendix A), including Maluku’s largest island (Seram), the smaller nearby island of Ambon and two islands within the Banda archipelago. Per island, we conducted four different manipulative trials on one single experimental site at approx. 250 m distance from the nearest maize field. All experiments were thus conducted in non-agricultural settings. Specifically, we assessed the extent of predation of canopy-exposed FAW egg masses, canopy- or soil-exposed larvae and soil-exposed FAW pupae. We equally assessed parasitism levels of the egg masses, larvae or pupae that were recovered in each trial. The different *S. frugiperda* life stages that were used for field experiments were obtained from a laboratory colony at the University of Pattimura in Ambon, Indonesia, which maintained maize plants as a natural host. Individual egg masses were obtained by allowing FAW moths to oviposit on paper strips that were hung in the rearing cages. The full set of predation trials was replicated two or three times per island over a six-month time period, covering the rainy season (May–August) and early to mid-stages of the dry season, i.e., September–October, October–November, respectively.

For the egg predation assay, we attached one single, newly deposited FAW egg mass to the leaf underside of a potted maize plant (V6-V10 phenological stage; [49]). Next, a 45 cm diameter, 90 cm high metal wire cage (mesh size 1 × 1 cm) was placed on top of each maize plant in order to exclude any vertebrate predators. Per field site (or island), a total of 10 plants were thus deployed at 50–100 m inter-plant distances and FAW egg masses were exposed to predation or parasitism over a 24 h period. Prior to its field exposure, we determined the exact number of eggs within each egg mass. After 24 h, egg masses were then retrieved from each caged plant, transferred to a field laboratory and the number of eggs or emerged first-instar larvae were counted. Given that (first-instar) FAW larvae engage in cannibalism, we used the proportion of intact eggs within a given egg mass as an imperfect proxy of egg predation. Next, egg masses were individualized in 6.5 cm diameter, 4.5 cm high screened cups and maintained at ambient temperature and humidity (27 °C, 80% relative humidity, RH) for min. two weeks. On a daily basis, cups were carefully examined and any eventual emerging parasitoids were recorded. Per site, egg predation assays were replicated twice. 

Canopy-exposed larval predation trials used the same set-up as above. Per caged maize plant, a total of ten thirds of instar FAW larvae were gently transferred onto different maize leaves and exposed to predation or parasitism over a 24 h period. Once this time period had elapsed, we carefully examined each plant, counted the number of surviving larvae and transferred those to the laboratory. Given the prominence of larval cannibalism and (occasional) escape, we used the proportion of surviving larvae on each plant as an imperfect proxy of larval predation. In the laboratory, larvae were individualized in 6.5 cm diameter, 4.5 cm high screened cups and maintained at ambient conditions (27 °C, 80% relative humidity, RH) for min. three weeks. During this time period, the eventual emergence of parasitoids was carefully assessed. These predation trials were replicated three times on Ambon and Seram islands, and twice on the Banda archipelago, i.e., Besar and Ai. 

For the soil-exposed larval predation trial, ten fifths of instar FAW larvae were attached to a 22 × 17 cm piece of cardboard using double-sided tape (^®^Tamago, PT Pesona Edukasi, Cilenggang Kota Tangerang Selatan, Banten, Indonesia). Per field site (or island), the thus affixed sentinel FAW larvae were exposed at seven locales at approx. 250 m distance from the nearest maize field. Sides of the cardboard piece were partly interred in the soil as to facilitate foraging by invertebrate predators. On top of each piece of cardboard, we further placed a 40 × 34 × 5 cm metal wire cage (mesh size 1 × 1 cm) to exclude any vertebrate predators. In close proximity to the exposed FAW larvae, we also deployed a pitfall trap to assess overall occurrence, abundance and species richness of ground-foraging invertebrate predators. Each pitfall trap consisted of a 6.5 cm diam., 6.6 cm high plastic cup that was put in the ground, filled with 5 cm of water and covered with 13.5 × 12.5 × 1.00 cm stryrofoam to exclude any rain. Sentinel larvae were exposed for 24 h; predation rates were recorded every two hours for the first six hours and a final assessment was performed after 24 h. Irrespective of an infrequent escape of exposed larvae, we used the proportion of surviving larvae over a given time span as an imperfect proxy of larval predation. Arthropod predators that were observed on the carboard or in the pitfall traps were counted and Identified to genus or morpho-species level. Larval predation trials were thus replicated three times on Ambon and Seram islands, and twice on Banda Besar and Banda Ai.

Lastly, the soil-exposed pupal predation trial entailed the exposure of five newly formed FAW pupae in a 9 cm Petri dish. The Petri dish was partly interred and filled with a 0.5–1 cm layer of coarsely sieved soil, which covered the sentinel pupae. Further, a 40 × 34 × 5 cm metal wire cage was placed on top of each Petri dish to exclude any vertebrate predators. Per field site (or island), ten different Petri dishes were thus spaced equidistantly at approx. 250 m from the nearest maize field. In the immediate vicinity of each Petri dish, we also deployed a pitfall trap, as described above. Sentinel pupae were exposed for a 24 h period, after which we counted the number of remaining pupae and transferred those to the laboratory. In the laboratory, pupae were placed within 6.5 cm diameter, 4.5 cm high screened cups and maintained at ambient conditions (27 °C, 80% RH) until parasitoid emergence. Arthropod predators that were found in the Petri dishes or in the pitfall traps were counted and identified to genus or morpho-species level. Pupal predation trials were replicated three times on Ambon and Seram islands, and twice on Banda Besar and Banda Ai.

For each type of predation trial, the apparent, stage-specific mortality was estimated as the proportion of individuals dying based on the number of exposed individuals. For each stage and island, apparent mortality figures were calculated by averaging predation rates across replicates. For soil-exposed larvae, we used the predation rates as recorded after 2 h to estimate apparent mortality. These figures then permitted developing a (truncated) life table for *S. frugiperda* on each given island [50]. Life tables reported the stage-by-stage cumulative mortality and the ensuing number of survivors across FAW life stages. Apparent mortality and cumulative mortality figures were both expressed as the proportion of individuals dying either in a given stage or also accounting for mortality in the preceding stages. 

For each of the above trials, egg, larval or pupal predation rates were compared between the 4 different islands. Given the large variability in predation rates between seasons, each replicate was analyzed separately. For the soil-exposed larval predation trial, we equally used a General Linear Model (GLM) to assess the effect of replicate (i.e., sampling event) and island on average longevity of the FAW larvae that survived 24 h or less. Larvae that survived beyond 24 h were included as censored data. Inter-group differences were determined using a Tukey HSD post hoc analysis. Kruskal–Wallis tests were also used to compare pitfall trapping data (i.e., predator abundance, richness) between the different island sites. Lastly, we used Pearson’s correlation analysis to relate site-by-site average predation rates for each of the different FAW life stages and realms, i.e., soil or canopy, and to also relate these with the respective predator abundance or richness measures. 

## 3. Results

### 3.1. CAM Natural Enemy Surveys

Across the five islands, field-level CAM abundance reached 26.7 ± 2.9 individuals (mean ± SE) per cassava tip and a maximum of 145.3 individuals per tip. In addition to the newly invasive *P. manihoti*, a number of other endemic or long-time invasive sap-feeding insects were recorded. The most prominent were the mealybugs *Pseudococcus jackbeardsleyi* Gimpel and Miller (3.3 ± 0.6 individuals per tip), *Ferrisia virgata* Cockerell (8.6 ± 2.5) and *Paracoccus marginatus* Williams and Granara de Willink (20.0 ± 4.1). Across sites, ants attained abundance levels of 5.5 ± 0.5 individuals per cassava tip. Meanwhile, a diverse community of generalist predators was associated with CAM, composed of lacewing larvae, adult or immature ladybeetles and spiders, at respective abundance levels of 0.2 ± 0.0, 0.3 ± 0.1 and 0.2 ± 0.1 individuals per tip. This generalist predator complex attained overall densities of 0.7 ± 0.2 individuals per tip. Field-level predator density correlated strongly with CAM density (Pearson’s r = 0.674, *p* < 0.01), but not with ant density (r = 0.199, *p* = 0.083). 

CAM density differed between cassava fields that were colonized by *A. lopezi* versus those that were not, attaining respective infestation levels of 20.4 ± 3.7 and 35.0 ± 4.4 individuals per tip (ANOVA, F_1,75_ = 6.692, *p* = 0.012). In *A. lopezi*-colonized fields, parasitism levels attained 37.5% ± 3.7% of mealybugs per cassava tip (range 0.1–81.25%). Meanwhile, resident primary parasitoids and hyperparasitoids acted at background levels in *A. lopezi*-colonized and uncolonized fields alike. Predators numerically responded to *P. manihoti* attack under certain settings; predator density was marginally and significantly regressed against CAM density in parasitoid-colonized fields (F_1,42_ = 3.834, *p* = 0.057, R^2^ = 0.084), but this was not the case in fields in which *A. lopezi* was absent (F_1,31_ = 0.250, *p* = 0.620). Further, predator/prey ratio was logarithmically regressed against CAM density in parasitoid-colonized fields (F_1,42_ = 37.921, *p* < 0.001, R^2^ = 0.474) as in those where *A. lopezi* was absent (F_1,30_ = 8.385, *p* = 0.007, R^2^ = 0.218). Even though predator density differed between fields with or without *A. lopezi* (ANOVA, F_1,75_ = 9.905, *p* = 0.002), this was not the case for predator/prey ratio (F_1,69_ = 0.957, *p* = 0.331). Lastly, predator/prey ratio was correlated with predator abundance in fields with and without *A. lopezi* (Spearman Rank ρ = 0.608, *p* < 0.001; ρ = 0.872, *p* < 0.001, respectively; Figure 1). 

Cassava mealybug density differed between the five islands (F_4,72_ = 6.103, *p* < 0.001), attaining the highest average density of 41.7–45.9 individuals per tip in Timor and Flores (Figure 2). Island size did not affect predator density for fields without *A. lopezi* (H = 4.256, *p* = 0.235) nor for those colonized by *A. lopezi* (H = 5.297, *p* = 0.071). Inter-island differences were recorded for predator/prey ratio in fields colonized by *A. lopezi* (H = 11.700, *p* = 0.003), but not in those without *A. lopezi* (H = 0.694, *p* = 0.875). For predator density, a final stepwise multiple regression model only included parasitoid presence (F_1,75_ = 9.905, *p* = 0.002, R^2^ = 0.117), excluding island and non-CAM density variables. Meanwhile, a final regression model for predator/prey ratio included *A. lopezi* presence and density of *P. manihoti* and co-occurring mealybug species (F_3,67_ = 17.302, *p* < 0.001, R^2^ = 0.437). Generalist predator abundance and predator/prey ratio are thus not directly determined by island size but by the presence or absence of an introduced parasitic wasp, the ensuing *P. manihoti* density and alternative mealybug prey numbers. Lastly, a final regression model for cassava mealybug density contained a density of co-occurring mealybugs, predator/prey ratio and *A. lopezi* presence (F_3,67_ = 24.092, *p* < 0.001, R^2^ = 0.519), but not island. Hence, neither invasive species’ success nor biotic resistance appear directly determined by island size. 

When examining insect census data at the plant (instead of field) level, notable similarities and differences were recorded. For CAM-affected plants only, predator density was significantly regressed against *P. manihoti* density in fields with and without *A. lopezi* (F_1,1075_ = 42.934, *p* < 0.001, R^2^ = 0.038; F_1,857_ = 19.825, *p* < 0.001, R^2^ = 0.046, respectively). As above, predator/prey ratio was logarithmically regressed against cassava mealybug density, but only in fields without *A. lopezi* (F_1,857_ = 40.945, *p* < 0.001, R^2^ = 0.046) and not in parasitoid-colonized ones (F_1,1045_ = 0.524, *p* = 0.469). Equally, predator/prey ratio and predator density exhibited strong correlation in fields with and without *A. lopezi* (Pearson’s r = 0.790, *p* < 0.001; r = 0.694, *p* < 0.001, respectively; Figure 1). Stepwise multiple regression showed how (plant-level) predator density is determined by *A. lopezi* presence and the densities of *P. manihoti* and other co-occurring mealybugs (F_3,1932_ = 77.745, *p* < 0.001, R^2^ = 0.108). A final model for predator/prey ratio retained island, predator density and *P. manihoti* density variables (F_3,1932_ = 929.435, *p* < 0.001, R^2^ = 0.591). Lastly, a final model for cassava mealybug included five variables, i.e., *A. lopezi* presence, density of predators and co-occurring mealybugs, island and the predator/prey ratio (F_5,1930_ = 115.884, *p* < 0.001, R^2^ = 0.231). Hence, in contrast to the field-level patterns, island size does exert a weak effect on invasive species success (i.e., *P. manihoti* abundance) and biotic resistance (i.e., predator/prey ratio) at the plant level. 

### 3.2. FAW Predation Trials

Across sites and sampling events, 49.2% of the eggs within the exposed FAW egg masses (average size of 145.6 ± 11.1 eggs per egg mass) were not recovered following a 24 h exposure on maize foliage. Egg predation rate was affected by seasonality, exhibiting inter-island differences during mid dry season (Kruskal–Wallis H = 8.103, *p* = 0.044), but not during the onset of the dry season (H = 5.996, *p* = 0.112; Figure 3A). Across sampling events, the lowest and highest egg predation rates were recorded in Ambon (24.1 ± 7.6%) and Banda Besar (66.4 ± 7.6%), respectively. Hence, the highest egg predation rate was logged on the second smallest island. 

Out of the early instar FAW larvae that were exposed on the maize canopy, 30.1 ± 2.1% were recovered after a 24 h period in all sampling runs and sites. Larval predation rates were affected by seasonality, exhibiting inter-island differences during the early dry season (H = 27.323, *p* < 0.001), but not during the mid dry season (H = 7.187, *p* = 0.066; Figure 3B). Across sampling events, the lowest and highest larval predation rates were recorded in Ambon (61.0 ± 7.2%) and Ai (99.0 ± 1.0%). Hence, highest predation rates were logged on the two smallest islands. 

Soil-exposed larvae experienced high levels of predation, which equaled 50.1 ± 4.8% after 2 h and 89.4 ± 2.6% after 24 h (Figure 4). Predation rates gradually increased over time in all sites, with the steepest increase in Besar and Ai islands. Survival rates equally reflected high predation rates on small islands; estimated survival duration was shortest on Besar (2.1 ± 0.0 h) and longest on Seram (18.1 ± 1.1 h). Survival duration was affected by both season and island (ANOVA, F_7,537_ = 48.403, *p* < 0.001, R^2^ = 0.387; Figure 4). Specifically, island, season and an island x season interaction term all exerted statistically significant effects on survival duration (F_3,537_ = 89.981, *p* < 0.001; F_1,537_ = 8.186, *p* = 0.004; F_3,537_ = 18.121, *p* < 0.001, respectively). Hence, survival of soil-exposed larvae was markedly lower in the two smallest islands. 

Out of all soil-exposed FAW pupae, 64.7 ± 4.3% were preyed upon over a 24 h time period (Figure 5). For the early and middle dry season alike, predation rates differed between islands (H = 9.921, *p* = 0.019; H = 32.163, *p* < 0.001, respectively). Across seasons, pupal predation rates ranged between 94–100% for Besar and 74–100% for Ai. Hence, soil-exposed pupae suffered consistently the highest predation rates in the two smallest islands. Across sites and sampling events, pitfall traps yielded 4.2 ± 1.8 individuals per trap, primarily composed of ants (89.0% trapped individuals) with occasional recordings of tiger beetles (Coleoptera: Cicindelidae; 6.9%), crickets (Orthoptera: Gryllidae; 2.4%) or spiders (Araneae; 1.4%). The number of putative predators caught in pitfall traps did not differ between islands (H = 1.422, *p* = 0.700). Lastly, upon transfer to the laboratory, none of the field-exposed FAW eggs, larvae or pupae yielded parasitoids. 

Based upon the above predation rates, apparent stage-specific mortality figures were calculated for FAW on each island (Table 1). A truncated life table shows how FAW survivorship beyond the pupal stage is 13- to 800-fold lower on the smaller islands of Besar and Ai, as compared to the largest island. This is also reflected in the cumulative mortality at the pupal stage, which attains a value of 0.9999 in the island of Besar. Hence, when departing from a hypothetical 10,000 eggs, one single FAW individual survives beyond the pupal stage in this particular island. 

## 4. Discussion

Understanding the relative contribution of biotic resistance is pivotal to managing invasive species preventatively. Further, in view of a rapid decline in ecosystem service providers such as insect predators or parasitoids [21,51,52], it is equally important to gauge to what extent biotic resistance intersects with on- and off-farm biodiversity. Here, we build upon the equilibrium theory of island biogeography (ETIB) to fill some of these knowledge gaps for two prominent invasive insect species that recently colonized Indonesia. First, observational assays with *P. manihoti* show how invasion success (i.e., mealybug density) differs between islands of varying size and a speciose complex of opportunistic predators attains high-abundance levels on mid-sized islands. Predators respond numerically to increasing mealybug densities at the level of individual plants, strongly underpinning biotic resistance (i.e., as inferred by predator/prey ratio), and their densities are shaped by resident mealybug populations. Yet, predator impacts—as mediated through island effects—on *P. manihoti* population regulation are overridden by those of an introduced parasitoid: across islands, *P. manihoti* densities are reduced by 43% more in parasitoid-colonized settings than in uncolonized ones. Second, manipulative assays with *S. frugiperda* uncover how resident predators inflict mortality levels up to 66.4%, 99.0%, 100.0% and 100.0% for its egg, larval and pupal stages on the plant canopy and soil, respectively. Though seasonally variable, predation rates are consistently high in presumed biodiversity-poor settings, i.e., small islands. As a result, survivorship of the invasive *S. frugiperda* is 13- to 800-fold lower on small versus large islands. Our work underlines how resident (invertebrate) predator communities deliver robust biotic resistance, even in presumably species-poor settings, and that invasive species mortality is further raised through the action of an introduced parasitic wasp. Yet, given the short experimental time-frame and the small set of islands in our study, caution needs to be taken when extrapolating our findings to other settings. 

Though islands are routinely thought to be exquisitely vulnerable to biotic invasion, this trend is far from universal [31,53]. Our work confirms how small oceanic islands, such as Banda Ai and Besar, are not inherently less resistant to invasive *S. frugiperda*, while cassava fields on the relatively small island of Flores, but not on Lombok, harbor abundant generalist predator communities, i.e., a core determinant of biotic resistance [54] (Crawley, 1986). As such, our findings are reminiscent of those by Gruner [22] and Beard et al. [55], in which native insectivores inflict substantial arthropod mortality on small islands. High abundance and activity patterns of generalist predators on small, biodiversity-poor islands, however, appear counter-intuitive, but this can be attributed to various factors. These include lessened degrees of intraguild predation or natural enemy distraction, e.g., by alternative prey or non-prey resources [56,57]. In more complex interaction networks, these phenomena are plausibly magnified and negatively impact predator foraging and feeding behavior [58], though co-occurring predators can raise biotic resistance under certain contexts [59]. Predation pressure can also be high because of fewer prey refuges, less (plant) resources for herbivorous prey, elevated risks of their stochastic extinction or lower carrying capacities of predators [60]. Generalist consumers, in particular, thrive within island settings, as these tend to be more mobile, less confined by certain habitats or affected by temporal resource availability [61]. More so, the island size below which they reach critical abundance is often further reduced by an island’s altitudinal elevation, habitat heterogeneity or productivity [60,62]. Island size effects are further modulated by specific traits of the taxon, climatic conditions or anthropogenic impact [63]. Hence, even on small, heterogeneous and highly productive islands, many generalist predators routinely offer reliable biological control and biotic resistance. In our assays, those effects are most evident when assessing *S. frugiperda* mortality at the soil versus canopy level, where they may be shaped by small-scale plant or habitat diversity and structural heterogeneity [64]. On the smallest islands in our experiment, predation rates of soil-exposed *S. frugiperda* pupae amply surpass those in its native range, e.g., with ants only removing 20% of experimentally deployed FAW pupae over a 24 h period in Nicaragua [65]. For canopy-level FAW larval predation, island level effects were apparent but comparatively weak and seasonally variable. Irrespective of the above, the extensive spread of *P. manihoti* along Indonesia’s island chain [34] reveals how resident predator communities do not fully repel invasions, but instead constrain population build-up and (ecological, socio-economic) impact [66].

Biodiversity provides vital insurance effects on overall ecosystem’s functioning, in which more species assure a stable supply of ecosystem goods and services under fluctuating environments [67,68]. Our work departed from a similar reasoning. As species-area relationships are steeper for higher trophic ranks [69], we assumed smaller islands to harbor fewer (specialist) consumers. It is unclear whether this assumption was met in our assays with *P. manihoti* and *S. frugiperda*: pitfall trapping failed to pick up inter-island differences in generalist predator abundance or richness, while CAM predator surveys lacked the necessary taxonomic resolution to compute richness or diversity indices. Also, as both species are recent invaders from the Neotropics, few if any specialist predators were thought to be present in Indonesia at the time of experimentation. For FAW, the high levels of larval and pupal mortality on small oceanic islands, i.e., Banda Ai or Besar, were likely attributed to (few) predatory ant species. This taxon of ubiquitous predators affects the populations of many arthropods, likely constituting some of the most voracious natural enemies in the tropics [70]. Even though ants are highly effective *S. frugiperda* predators [71], they are routinely ignored in biological control research. As ants act as intermediate trophic level super-generalists, the presence of just a few ant species in small islands can have important implications for food-web structure and an ecosystem’s functionality [62]. For CAM, a speciose complex of generalist predators, composed of lacewings, ladybirds and spiders, reached abundance levels of 0.7 individuals per cassava tip across the five islands. Cassava grown under agroforestry arrangements in the island of Flores harbored an average of 3.2 (primarily orb-weaver) spiders per plant at the field-level. Tens of ladybird species are associated with *P. manihoti* in its invasive range [38], but they are considered ineffective in regulating populations of insects with short generation times such as mealybugs [72]. Spiders are equally voracious predators of many crop-feeding herbivores, but different spider genera and life stages exhibit varying propensity for either intraguild or pest predation [73]. Web-builders and spiders with ambush hunting strategies may also be less effective predators of sessile insects such as mealybugs. Lastly, resident generalists can exhibit preferences for native prey, which may be outspoken early in the invasion process [74] or when a new invader sequesters plant toxins, e.g., as may be the case for *P. manihoti* [75]. Hence, for generalist ladybirds, spiders or ants alike, the nature and relative importance of (native, invasive) pest–predator and predator–predator interactions carries major repercussions for ecosystem functioning [76]. Clearly, without a full picture of the trophic ecology of the (FAW, CAM) predator guild in the various island settings (e.g., [77]), it is challenging to draw meaningful conclusions on biodiversity x ecosystem functioning within our experimental contexts. This also underscores the value of mechanistic population-level studies that account for multi-level food webs: a central premise of the trophic theory of island biogeography (TTIB; [62,78]). It equally underlines a need to incorporate predator life history parameters (e.g., functional or numerical response) and prey switching in biotic resistance estimates [79].

Predators inflict substantial levels of FAW or CAM mortality, i.e., as measured or inferred through our assays. In mealybug-invaded cassava fields, the generalist predator complex responded numerically to *P. manihoti* density, particularly at a plant level. This is in line with findings from European wheat fields or Californian walnut orchards, where polyphagous ground beetles, rove beetles or ladybeetles aggregate to patches of homopteran prey [48,80,81]. However, as simple correlations between prey and predator densities are not necessarily causal [72], one cannot infer that highly abundant generalist predators in the mid-sized islands of Timor or Flores actually regulate CAM populations. Such is also mirrored in the predator/prey ratio metric; a proxy of biotic resistance that showed no inter-island differences. This metric sharply declined with increasing *P. manihoti* density, which may be indicative of predators’ rapid saturation by food abundance. Hence, as also observed for both gypsy moth and winter moth in North America [41,42], generalist predators exert the strongest top-down control, or attain the highest predator/prey ratio, in fields with low-pest numbers. This rapid decline in biological control activity with mealybug density is clear in most (but not all) fields where the introduced specialist parasitoid *A. lopezi* was absent. Fields with unusually strong biotic resistance are possibly marked by more speciose predator communities, though this remains to be verified. Also, in order to ascertain the exact extent of predator-mediated population suppression, we need to step beyond our current ‘snap-shot’ survey and conduct season-long combined assessments of mealybug and predator populations. In *A. lopezi*-colonized fields, biological control activity proved comparatively stable across a range of CAM densities and strongly responsive to predator density, especially at plant level (e.g., [82]). Meanwhile, a strong regulatory effect of *A. lopezi* (as demonstrated through multi-year population studies; [34]) is evident in the consistently high level of *P. manihoti* suppression across islands and its cascading impacts on predator abundance. Across sites, *A. lopezi* attained parasitism levels of 37.5% of mealybugs, but was likely in the initial stages of establishment in the island of Lombok. These levels are well beyond maximum parasitism rates of effective biological control, i.e., 33–36% [83]. Hence, in a similar way for winter moth, walnut aphid or soybean aphid in North America [42,48,84], one single host-specific parasitoid appears to work in tandem with a diverse community of generalist predators to regulate invasive herbivore populations.

By drawing upon observational and manipulative studies, our work unveils how generalist (invertebrate) predators contribute to biotic resistance even in biodiversity-poor settings, i.e., small oceanic islands. Ground-foraging predators such as ants or crickets, and foliage-dwellers such as ladybirds, lacewings or spiders inflict substantial (measured or inferred) mortality among two prominent invasive crop pests. While biotic resistance is ubiquitous along a chain of Indonesian islands of varying size [85], its magnitude differs markedly between island contexts, seasons and ecological realms, i.e., plant canopy vs. soil surface. Though biodiversity may not directly dictate agro-ecosystem robustness (i.e., the capacity to withstand novel shocks such as invasive pest attack), it most certainly determines resilience, i.e., the ability to bounce back from disturbance. Insect biodiversity per se is likely only one of several (biotic, abiotic) determinants of biotic resistance [86]. Thus, upholding the ecological integrity or overall ecosystem health may be central to an effective mitigation of invasive species. For species-poor farmland ecosystems in particular, this entails the removal of unnecessary disturbances, e.g., pesticide applications in parallel with efforts to add in (plant) diversity, incorporate ecological infrastructures or embed fields within heterogeneous landscape mosaics [18,87,88]. Given the unremitting Anthropocene defaunation and ecosystem degradation [89,90], our island experiments underscore how a deliberate conservation of predatory arthropods can bolster the resilience of biodiversity-poor farmland ecosystems. Given that chemical pesticides pose key mortality factors for those beneficial biota, non-chemical alternatives are to be consciously prioritized for crop protection and invasive species mitigation in and beyond fragile island systems.

## Figures and Tables

**Figure 1 plants-12-03304-f001:**
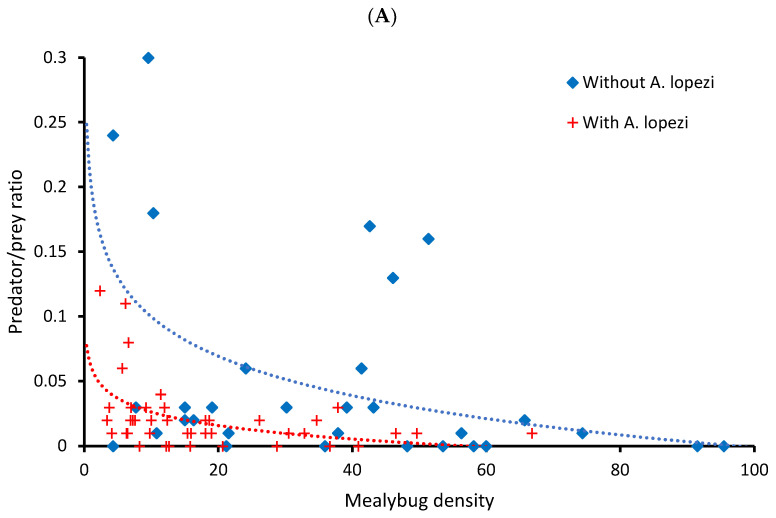
Biological control activity relates to *P. manihoti* infestation pressure and predator density in cassava fields with or without an introduced parasitoid. In the panel (**A**), a predator/prey ratio is obtained by dividing the field-level density of all predators by that of *P. manihoti*. (Marginally) significant linear or logarithmic regression curves are shown. In the panel (**B**), correlation trends are shown between log(x + 1) transformed predator/prey ratio and predator density, as recorded at the plant- or field-level. Both predator and mealybug density represent the number of individuals per plant. Patterns are plotted for settings with *A. lopezi* (red) and those without (blue). In the panel (**B**), full and dashed lines mirror correlation trends at the plant or field level, respectively. For visualization purposes, the X axis of the upper panel was truncated at 100 mealybugs/plant and one predator/prey ratio outlier removed from panel (**A**) and panel (**B**), without affecting the statistical significance of regression curves. All statistical details are provided in the main text.

**Figure 2 plants-12-03304-f002:**
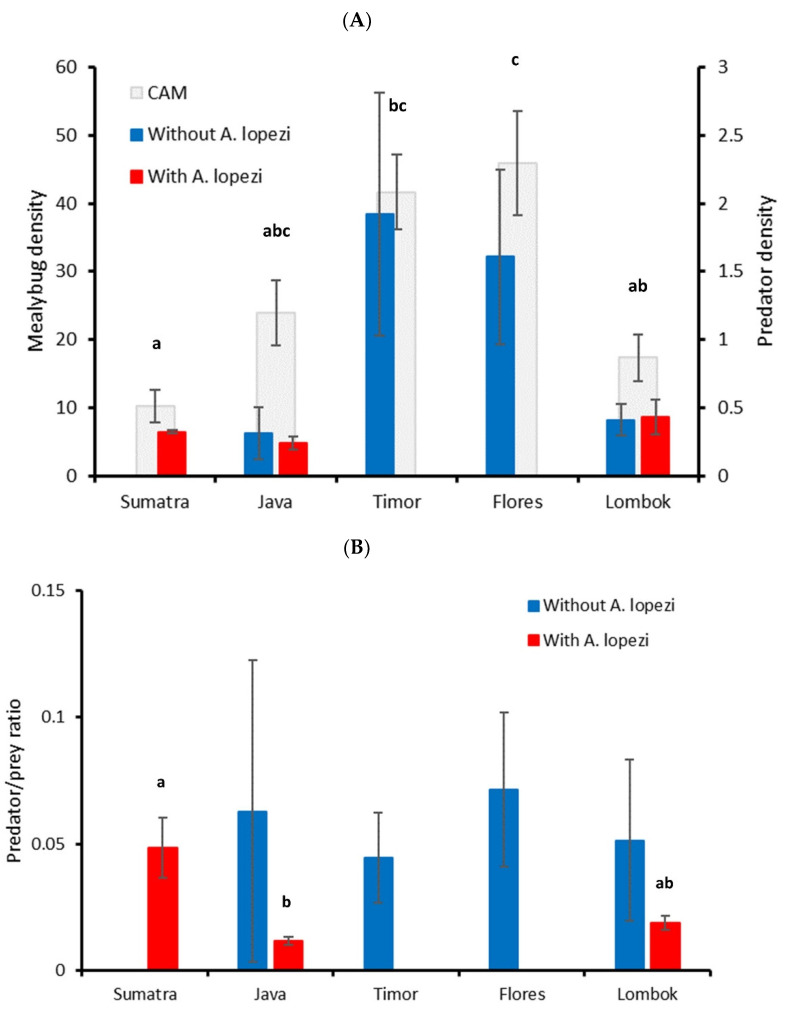
Inter-island differences in invasive mealybug density, predator density and predator/prey ratio for cassava fields with differing colonization by the introduced parasitoid *Anagyrus lopezi*. The panel (**A**) shows inter-island differences in cassava mealybug (grey bars; main Y axis) or total predator abundance (red or blue bars; secondary Y axis) per plant, for fields with or without *A. lopezi*. In the panel (**B**), depicting field-level predator/prey ratio, one outlier is removed for data visualization purposes. Islands are ranked from large to small in size. Statistical differences are indicated by different letters above each bar (ANOVA or Kruskal–Wallis, *p* < 0.05) and are only shown for those variables that exhibit inter-island differences. Statistical details are provided in the main text.

**Figure 3 plants-12-03304-f003:**
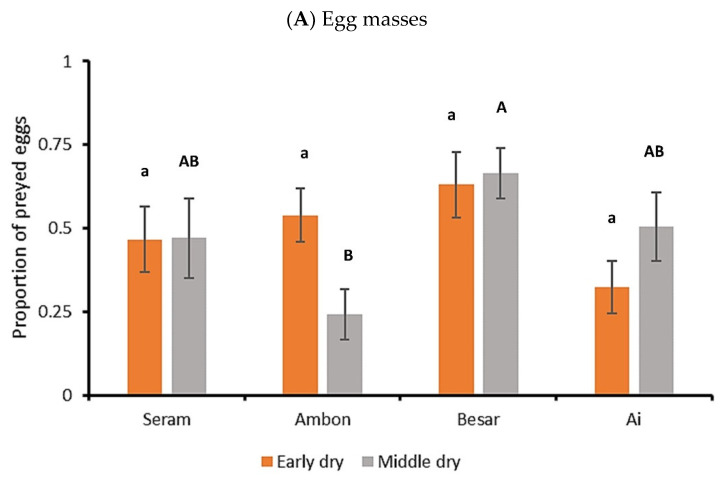
Predation rates of canopy-exposed FAW egg masses (**A**) and sentinel larvae (**B**) in four Indonesian islands of varying size. The graphs depict the average proportion (±SE) of FAW eggs or larvae that are not recovered after 24 h, following their exposure on potted maize plants. Data are shown for experiments conducted during the rainy, early dry and middle dry season. In panel (**B**), data are shown for three separate sampling events on Seram and Ambon, and two sampling events on Besar and Ai islands. Statistical differences are indicated by different (capital, lowercase) letters above each bar (Kruskal–Wallis, *p* < 0.05). Statistical data are provided in the main text.

**Figure 4 plants-12-03304-f004:**
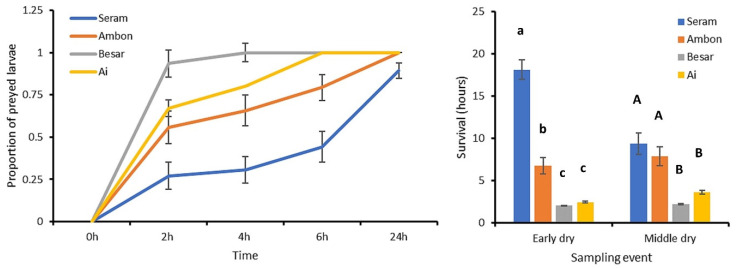
Predation over time of soil-exposed FAW sentinel larvae in four Indonesian islands of varying size. The left panel depicts the average proportion (±SE) of sentinel larvae that are preyed upon over four specific periods of time, i.e., 2 h, 4 h, 6 h and 24 h. Predation patterns are plotted for Seram, Ambon, Besar and Ai (from large to small in size). The right panel shows average (±SE) survival times for two different sampling events. Larvae that survive beyond 24 h are included as censored data in this graph. Statistical differences are indicated by different (capital, lowercase) letters above each bar (ANOVA, Tukey HSD, *p* < 0.05). Statistical data are provided in the main text.

**Figure 5 plants-12-03304-f005:**
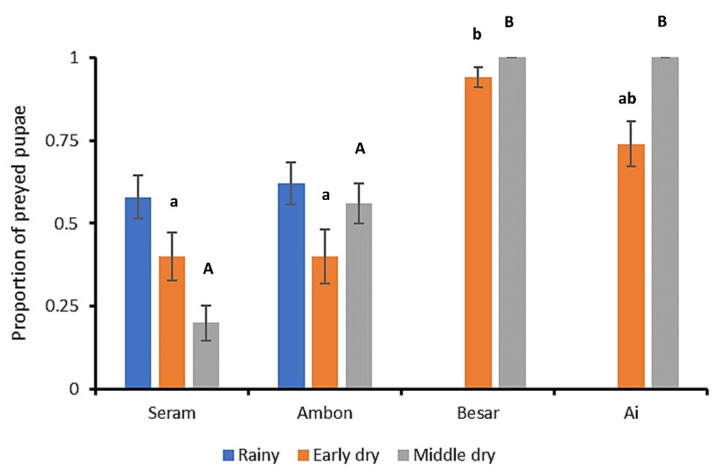
Predation rates of soil-exposed FAW pupae in four Indonesian islands of varying size. The graph depicts the average proportion (±SE) of FAW pupae (out of 10) that are not recovered after 24 h, following exposure on the soil surface. Data are shown for three separate sampling events on Seram and Ambon, and two sampling events on Besar and Ai islands. Statistical differences are indicated by different (capital, lowercase) letters above each bar (Kruskal–Wallis, *p* < 0.05). Statistical data are provided in the main text.

**Table 1 plants-12-03304-t001:** Truncated life table for *Spodoptera frugiperda* on four Indonesian islands of varying size and geographical isolation. Cumulative mortality estimates are exclusively based on stage-specific apparent mortality levels that were recorded during (canopy- and/or soil-exposed) egg, larval and pupal predation trials. This imperfect method does not take into account other (e.g., development, rain dislodgement, parasitism or cannibalism-related) mortality factors nor developmental duration of each stage. We depart from 10,000 eggs to estimate the number of survivors entering a given developmental stage. Islands are ranked from large (top) to small (bottom) in size.

Parameter	Island	Development Stage				
		Egg	Early Instar Larva	Late-Instar Larva	Pupa	Adult
Cumulative mortality	Seram	-	0.818	0.867	0.919	-
	Ambon	-	0.768	0.897	0.951	-
	Besar	-	0.948	0.997	0.999	-
	Ai	-	0.856	0.953	0.994	-
Number of individuals entering stage	Seram	10,000	5320	1825	1330	807
	Ambon	10,000	6100	2318	1027	486
	Besar	10,000	3530	522	33	1
	Ai	10,000	5860	1435	472	61

## Data Availability

The underlying data of this manuscript are available upon reasonable request from the authors.

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
