# Peer review of "Generalist Predators Shape Biotic Resistance along a Tropical Island Chain"

_plants, 2023, doi:10.3390/plants12183304_

Round 1
Reviewer 1 Report
The current MS of plants-2600832 by Kris AGW et al., showed us a very interesting work on a chain of 9 isolated islands to quantify predator-mediated biotic resistance against the fall armyworm Spodoptera frugiperda and the cassava mealybug Phenacoccus manihoti. Glad to see isolated islands work on the massive scale, and measures of predator biocontrol. This 9 isolated islands evaluation provides interesting example data to shown that resident predators inflict apparent mortality rates for S. frugiperda. For P. manihoti, a diverse set of generalist predators aggregates on mealybug-infested plants – attaining max. (field-level) abundance levels of 1.0, 8.0 and 3.2 individuals per plant, respectively. The current work revealed that the generalist predators are key determinants of biotic resistance, exerting important levels of pest biocontrol even in biodiversity-poor settings. This topic is rare, very impressive, and it will certainly influence related researcher to follow. Furthermore, the writing does a good job of putting this region in a global context and raising issues that extend far beyond its borders.
Recommendation
Acceptable after minor revision
Minor comments
1. Page 3, 2.1 CAM natural enemy surveys, and Page 4, 2.2 FAW predation trials, I suggest to add a more clearly GIS map in the main text and combined with the Supplementary Table 1, then to show the different island settings Indonesia.
2. I suggest to rearrange the logic of the currents MS, i.e. in the abstract and results part, it were arranged by predators on FAM and then on CAM, however, in the Materials & Methods, 2.1 CAM natural enemy surveys, 2.2 FAW predation trials, and all of the data analysis was merged in FAW and CAM, respectively. How about to isolated the data analysis into 2.3?
3. Line 223-230:In the Canopy-exposed larval predation trials, the ten 3rd larvae were gently transferred onto the maize leaves and exposed for 24 hours period. It should be reminder, maybe some of the larvae escape by itself? Then, it eventually counted in the predation or not? This maybe also the same for the 5th instar fixed on the cardboard using double-sided tape (Line 234-240).
Author Response
The current MS of plants-2600832 by Kris AGW et al., showed us a very interesting work on a chain of 9 isolated islands to quantify predator-mediated biotic resistance against the fall armyworm Spodoptera frugiperda and the cassava mealybug Phenacoccus manihoti. Glad to see isolated islands work on the massive scale, and measures of predator biocontrol. This 9 isolated islands evaluation provides interesting example data to shown that resident predators inflict apparent mortality rates for S. frugiperda. For P. manihoti, a diverse set of generalist predators aggregates on mealybug-infested plants – attaining max. (field-level) abundance levels of 1.0, 8.0 and 3.2 individuals per plant, respectively. The current work revealed that the generalist predators are key determinants of biotic resistance, exerting important levels of pest biocontrol even in biodiversity-poor settings. This topic is rare, very impressive, and it will certainly influence related researcher to follow. Furthermore, the writing does a good job of putting this region in a global context and raising issues that extend far beyond its borders.
Response: We are grateful to reviewer #3 for seeing value in our work, and for providing several useful suggestions.
- Page 3, 2.1 CAM natural enemy surveys, and Page 4, 2.2 FAW predation trials, I suggest to add a more clearly GIS map in the main text and combined with the Supplementary Table 1, then to show the different island settings Indonesia.
Response: We personally feel that - with five (multi-panel) figures and one table- already many display items are featured in our manuscript. We thus prefer not to include an additional GIS map.
- I suggest to rearrange the logic of the currents MS, i.e. in the abstract and results part, it were arranged by predators on FAM and then on CAM, however, in the Materials & Methods, 2.1 CAM natural enemy surveys, 2.2 FAW predation trials, and all of the data analysis was merged in FAW and CAM, respectively. How about to isolated the data analysis into 2.3?
Response: As suggested, we have now re-arranged the order of presentation of FAW and CAM-related experiments in the abstract. As different statistical tests were used for either type of study, we prefer to elaborate further on the data analysis aspects within the respective paragraphs – instead of merging all these elements into a stand-alone paragraph. Given the complexity of our study, we are confident that this will benefit future readers and avoid unnecessary confusion.
- Line 223-230:In the Canopy-exposed larval predation trials, the ten 3rd larvae were gently transferred onto the maize leaves and exposed for 24 hours period. It should be reminder, maybe some of the larvae escape by itself? Then, it eventually counted in the predation or not? This maybe also the same for the 5th instar fixed on the cardboard using double-sided tape (Line 234-240).
Response: We are thankful to reviewer #3 for this very good observation. We have now amended the text as such: “Irrespective of an infrequent escape of exposed larvae, we used the proportion of surviving larvae over a given time span as an imperfect proxy of larval predation (line 261-263)” and “Given the prominence of larval cannibalism and (occasional) escape, we used the proportion… (line 244)”.
Reviewer 2 Report
In this manuscript, Wyckhuys et al present a valuable and well-structured investigation into the interplay between biodiversity and ecosystem function on a chain of Indonesian islands. It highlights the significance of these islands as unique environments for studying predator-prey interactions, specifically focusing on the fall armyworm and the cassava mealybug.
The research demonstrates the role of resident predators, such as soil-dwelling ants, in significantly reducing the survivorship of the fall armyworm on small islands compared to larger ones. Additionally, it revealed the presence of diverse generalist predators regulating the cassava mealybug population.
Their overall conclusion emphasizes the importance of conserving generalist predators in agricultural settings to enhance the resilience of production ecosystems, especially in the face of biodiversity loss and the introduction of alien species.
The study shows how even in presumed biodiversity-poor settings, such as small oceanic islands, generalist predators play a pivotal role in inflicting substantial mortality on invasive species. This highlights the potential for natural pest control in agricultural systems, underscoring the importance of conserving predator communities and reducing pesticide use to maintain ecosystem health and resilience.
All is well explained in this manuscript. I do not have concerns with it, therefore I recommend to continue the process towards publication.
Author Response
We wish to thank reviewer #1 for his/her positive assessment of our manuscript.
Reviewer 3 Report
According to the Theory of Island Biogeography, components of the structure of biological communities like species richness, can be described as functions of island size. The present study is an attempt to reveal if ecological processes such as species interactions (like predation) can similarly be related to island size. This is a formidable task, and the results give only vague indications. The study tests how “biotic resistance” (i.e. predators and parasitoids) affects the establishment of two recent invasive agricultural pest species on Indonesian islands of varying size. One pest species were studied on 4 islands, the other on 5 islands. The limitation on island numbers prevents rigorous statistical relationships of results to island size. Thus, conclusions such as “predators attains high abundance levels on mid-sized islands”, “predation rates are consistently high in presumed biodiversity-poor settings i.e., small islands” seem much too generalized; the reader is not convinced that a different set of island would reveal the same results.
Another problem is whether, or to what extent, the Theory of Island Biogeography can be extended to apply to the agricultural fields. As habitats, these are greatly simplified due to agricultural practices (were the used fields sprayed?), and the assumption that the natural predator community of agricultural fields is richer on large islands (cf. citation above) is not obvious. Only a minor part of the island’s complement of predators will be represented in the fields. Furthermore, predator richness is not measured; the study uses total numbers of predators.
It is concluded that “generalist predators are key determinants of biotic resistance” (Abstract). When biotic resistance is defined as the ratio of number of predators to number of pests, there are only these two determinants, and both are “key”. The problem is that a structural characteristic (predator/prey ratio) is interpreted as a dynamic one, i.e. as indicating an ecological process.
The strongest part of the study is the experimental tests of predation on various stages of the fall armyworm. These tests seem to demonstrate that predation is a strong interaction in the agricultural system, but the relationship to island size is doubtful.
Author Response
Response: We are grateful to reviewer #2 for these valuable, constructive comments. Below, we provide a point-by-point response on how our manuscript has been adapted to meet these comments. We are confident that these adaptations raise the overall quality of our manuscript; we are equally hopeful that they will meet the reviewer’s expectations.
According to the Theory of Island Biogeography, components of the structure of biological communities like species richness, can be described as functions of island size. The present study is an attempt to reveal if ecological processes such as species interactions (like predation) can similarly be related to island size. This is a formidable task, and the results give only vague indications. The study tests how “biotic resistance” (i.e. predators and parasitoids) affects the establishment of two recent invasive agricultural pest species on Indonesian islands of varying size. One pest species were studied on 4 islands, the other on 5 islands. The limitation on island numbers prevents rigorous statistical relationships of results to island size. Thus, conclusions such as “predators attains high abundance levels on mid-sized islands”, “predation rates are consistently high in presumed biodiversity-poor settings i.e., small islands” seem much too generalized; the reader is not convinced that a different set of island would reveal the same results.
Response: We fully concur with reviewer #1 that the small set of islands and the short experimental time-frame limits constitute important limitations. To emphasize these limitations further, we now include in line 418 “Yet, given the short experimental time-frame and the small set of islands in our study, caution needs to be taken when extrapolating our findings to other settings.”
Another problem is whether, or to what extent, the Theory of Island Biogeography can be extended to apply to the agricultural fields. As habitats, these are greatly simplified due to agricultural practices (were the used fields sprayed?), and the assumption that the natural predator community of agricultural fields is richer on large islands (cf. citation above) is not obvious. Only a minor part of the island’s complement of predators will be represented in the fields. Furthermore, predator richness is not measured; the study uses total numbers of predators.
Response: We recognize that some details on the methodology were missing. These elements have now been filled in as such: For the FAW predation trials, we clarify that “All experiments were thus conducted in non-agricultural settings.” (line 217). For the mealybug predation trials, we indicate that “Fields were managed by individual farmers using locally-prevailing practices; as cassava is a low-value, long-season crop that is primarily cultivated by resource-poor farmers (especially) in Flores, Timor and Lombok, pesticide use is rare.” (line 164-167).
It is concluded that “generalist predators are key determinants of biotic resistance” (Abstract). When biotic resistance is defined as the ratio of number of predators to number of pests, there are only these two determinants, and both are “key”. The problem is that a structural characteristic (predator/prey ratio) is interpreted as a dynamic one, i.e. as indicating an ecological process.
Response: We agree with reviewer #2 on the incorrect wording in the Abstract. We have now amended this sentence to “Generalist predators determine biotic resistance…”.
The strongest part of the study is the experimental tests of predation on various stages of the fall armyworm. These tests seem to demonstrate that predation is a strong interaction in the agricultural system, but the relationship to island size is doubtful.
Response: We wish to thank reviewer #2 for pointing out this important finding.